# DBA-YOLO: A Dense Target Detection Model Based on Lightweight Neural Networks

**DOI:** 10.3390/jimaging11100345

**Published:** 2025-10-04

**Authors:** Zhiyong He, Jiahong Yang, Hongtian Ning, Chengxuan Li, Qiang Tang

**Affiliations:** 1College of Information Science and Engineering, Hunan Normal University, Changsha 410000, China; 202320294080@hunnu.edu.cn (Z.H.); jhyang3668@hunnu.edu.cn (J.Y.); 202370294147@hunnu.edu.cn (H.N.); 202320294076@hunnu.edu.cn (C.L.); 2College of Engineering and Design, Hunan Normal University, Changsha 410000, China

**Keywords:** object detection, multi-scale, feature fusion, small-size objects, detection head

## Abstract

Current deep learning-based dense target detection models face dual challenges in industrial scenarios: high computational complexity leading to insufficient inference efficiency on mobile devices, and missed/false detections caused by dense small targets, high inter-class similarity, and complex background interference. To address these issues, this paper proposes DBA-YOLO, a lightweight model based on YOLOv10, which significantly reduces computational complexity through model compression and algorithm optimization while maintaining high accuracy. Key improvements include the following: (1) a C2f PA module for enhanced feature extraction, (2) a parameter-refined BIMAFPN neck structure to improve small target detection, and (3) a DyDHead module integrating scale, space, and task awareness for spatial feature weighting. To validate DBA-YOLO, we constructed a real-world dataset from cigarette package images. Experiments on SKU-110K and our dataset show that DBA-YOLO achieves 91.3% detection accuracy (1.4% higher than baseline), with mAP and mAP75 improvements of 2–3%. Additionally, the model reduces parameters by 3.6%, balancing efficiency and performance for resource-constrained devices.

## 1. Introduction

In recent years, deep learning has markedly improved the efficiency of object detection, delivering superior performance and notable success over traditional methods. This paradigm shift has shown strength across diverse areas such as autonomous driving [1], face recognition [2], and text recognition [3], and it has become integral to many downstream applications. Despite the revolutionary impact of deep learning in detection, challenges remain; for example, state-of-the-art detectors are often computationally intensive. This demand creates a significant barrier to deployment on resource-constrained devices, a dilemma amplified by the rapid proliferation of mobile platforms such as law-enforcement recorders and fixed-wing UAVs [4]. Against this backdrop, we target dense-object detection on edge devices and emphasize a design orientation that simultaneously pursues high accuracy and high efficiency.

In real-world settings, dense-object detection is especially challenging in retail shelf displays [5], UAV imagery [6], and the cigarette-package images showcased in Figure 1. We focus on methodological advances for dense detection—multi-scale feature fusion, attention-guided aggregation under crowding/occlusion, and real-time decoding—and cite application studies to indicate cross-scene applicability across merchandise/crowd scenarios and UAV-based image analysis [7,8,9]. These scenarios typically contain many similar or identical instances packed at high density, often with occlusions. As density increases, detectors must preserve fine-grained localization cues and remain robust to severe overlaps and complex background clutter.

Convolutional Neural Networks (CNNs) have demonstrated strong capabilities in image understanding and feature representation [10]. By extracting key visual features, CNN-based models can significantly improve detection performance. Dense-object detectors are broadly categorized into two paradigms: two-stage and single-stage. Two-stage detectors follow the cascade of “region proposal—feature extraction—classification”, typified by the R-CNN family; although accurate, their reliance on proposal generation and feature resampling constrains real-time performance in dense scenes. In contrast, single-stage detectors adopt dense prediction and directly perform classification and regression without pre-generated proposals. In practice, detection systems are often deployed on edge/mobile devices with limited computing power. (Zaidi et al., 2022) [11]. Large models struggle under such constraints, so lightweight design (e.g., efficient operators and compact backbones) and model compression (pruning/quantization) have become key directions. Although these techniques reduce complexity, maintaining accuracy in dense small-object settings is still challenging because aggressive compression can attenuate early high-resolution cues and destabilize localization. To mitigate this, handcrafted lightweight networks such as MobileNet (Sandler et al., 2018) [12], ShuffleNet (Ma et al., 2018) [13], and GhostNet (Han et al., 2020) [14] optimize convolutions to build efficient architectures suitable for constrained devices. We keep the exposition method-centric and summarize application works briefly to avoid drifting off theme.

In dense retail shelves and cigarette package scenes, objects are tightly arranged with high spatial proximity and similar appearance, making boundary discrimination difficult; the coexistence of multiple scales in a single image further increases the challenge. Consequently, directly transferring lightweight models designed for low-density settings can be suboptimal. Our design adheres to three principles: (i) preserve and exploit early high-resolution information; (ii) perform direction-balanced and cost-aware multi-scale fusion; and (iii) employ attention-guided, foreground-prioritized feature selection to withstand crowding and occlusion. As a representative real-time family, the YOLO series aims to balance speed and accuracy. Given YOLOv10’s scalable architecture and favorable accuracy/latency trade-off (Wang et al., 2024) [15], we take YOLOv10 as the baseline and refine the architecture, feature-extraction strategies, and localization head to meet the stringent requirements of dense scenes on edge devices. In implementation and evaluation, we favor compact, synergistic designs over additive stacking and validate them quantitatively on mainstream benchmarks.

We start from YOLOv10 as the base model. To harvest more informative features, we propose a plug-and-play neck—the weighted bidirectional multi-branch assisted feature pyramid network (BIMAFPN). In BIMAFPN, BiSAF (shallow assisted fusion) maintains shallow backbone information via bidirectional connections to enhance small-object detection, and BiAAF (advanced assisted fusion) enriches output-layer gradients via multidirectional connections; meanwhile, BiFPN improves efficiency and accuracy through bidirectional cross-scale connections with weighted fusion. Second, because feature regions differ in information content and foreground cues should be emphasized, we introduce the C2f PA module that augments C2f with attention, enabling adaptive reweighting by importance to improve feature selection. Finally, for accurate detection in complex dense scenes, we design the attention-enhanced head DyDHead, which builds on the YOLOv10 head with a dynamic mechanism that combines Dynamic Deformable Convolution (DyDCNv3) and multiple attentions to strengthen feature-map representation. We validate our approach on SKU110K and a curated cigarette package dataset. In addition, to probe cross-scene generalization, we conduct a small-scale comparative test on the VisDrone dataset with several representative models as supplementary evidence.

Our contributions are as follows:We propose an improved C2f PA module in the backbone that adaptively adjusts feature weights according to their importance, thereby improving feature extraction. Unlike simply stacking generic attention, C2f PA prioritizes foreground information and selectively enhances shallow, fine-grained cues, targeting the early-stage erasure problem common to dense small objects.We present DyDHead, an improved detection head built upon YOLOv10 that integrates novel dynamic convolution, adaptive feature enhancement, and multi-scale semantic awareness for more accurate target characterization in complex scenes, yielding significant performance gains. DyDHead combines dynamic deformable sampling with hierarchical attention to alleviate localization instability under occlusion/overlap, while a lightweight path design keeps the extra overhead controlled.We propose BIMAFPN, a weighted bidirectional multi-branch assisted FPN that combines BiFPN with auxiliary branches for richer interaction and fusion. BiSAF preserves shallow information for small-object sensitivity; BiAAF enriches output-layer gradients via multidirectional links; and BiFPN provides learnable, bidirectional cross-scale fusion to improve efficiency and accuracy while reducing parameters. Unlike directly concatenating a generic neck, BIMAFPN employs a “shallow-fidelity + high-level gain” dual-assist pathway explicitly tailored to dense small objects and supplies features matched to the detection head.We build a practical cigarette package dataset for testing, comprising 1073 images and 50,173 instances at 960 × 1280 resolution. As an application-neutral dense benchmark, it supports reproducible evaluation for methods targeting crowding and small objects. We plan to expand it to 3000 real images and, with augmentation, to 5000 images and 200,000 instances for public release.

## 2. Related Work

### 2.1. Real-Time Dense Small-Object Detection

Research on real-time, dense small-object detection has progressed along two lines and is gradually converging. On the YOLO line, designs emphasize lightweight multi-scale fusion and very low end-to-end latency: early versions broadened scale coverage with SPP and PANet [16,17,18]. YOLOv8 [19] adopts an anchor-free, decoupled head to stabilize optimization. YOLOv10 [15] further restructures the pipeline for edge deployment under tight latency budgets, achieving fewer parameters and lower FLOPs at comparable accuracy while maintaining stable throughput. The subsequent YOLOv11/12 [20,21] introduce attention-centric refinements in the backbone/neck while preserving real-time performance.

In parallel, transformer-based detectors reformulate detection as one-to-one set prediction and use attention to model global dependencies: DETR [22] establishes the end-to-end paradigm via Hungarian matching; Deformable DETR [23] accelerates convergence and improves small-object performance through multi-scale deformable attention; and RT-DETR/RT-DETRv2 [24] advance these ideas toward real-time, deployable regimes with practical attention/sampling strategies and training “freebies”. In addition, DEIM [25] introduces dynamic early inference/early exit into end-to-end detection—using confidence/entropy-based or token-pruning adaptive inference to reduce latency and energy while largely preserving accuracy.

A complementary strand on dense/crowded scenes directly addresses overlap and tiny objects, including crowd-aware de-duplication, one-to-one assignment in fully convolutional detectors, and sparse/dense query mechanisms for high-resolution small objects (e.g., CrowdDet, DeFCN/POTO, QueryDet, DDQ) [26,27,28,29]; building on this, the dynamic inference direction (e.g., DynamicDet [30]) further employs a detection-oriented dynamic architecture with an adaptive router to select inference paths on demand across multi-scale features, and it uses a detection-loss-based exit criterion to realize variable-speed inference, thereby covering a broader accuracy–speed trade-off with a single model. Taken together, three practical principles have emerged for dense scenarios: preserve early high-resolution cues, perform cost-aware and direction-balanced multi-scale fusion, and leverage attention-guided foreground-prioritized feature selection to remain robust under crowding and occlusion.

Positioning and baseline choice. Given our target scenarios with high object density and strict latency/memory budgets, we adopt YOLOv10 as the primary baseline: it offers an excellent speed–accuracy–latency trade-off, lower parameter and FLOP budgets suitable for on-device deployment, and stable throughput under edge constraints while being structurally compatible with lightweight multi-scale fusion necks and heads.

### 2.2. Multi-Scale Feature Fusion

Multi-scale feature fusion aggregates features at different resolutions to enable cross-scale object detection. FPN [31] adopts a top-down pathway with lateral connections to inject high-level semantics into low-level maps, while PANet [32] complements this with a bottom-up path that strengthens localization and shortens the information flow between shallow and deep layers. Building on these ideas, BiFPN [33] introduces repeated bidirectional fusion with learnable non-negative weights and lightweight refinement, normalizing per-scale contributions and markedly improving the accuracy–efficiency trade-off.

Regarding fusion quality and selectivity, AugFPN [34] narrows semantic gaps across levels via enhancement and attention modules, whereas CE-FPN [35] mitigates channel-reduction loss and cross-scale aliasing through sub-pixel fusion and channel-attention guidance. To further alleviate semantic degradation across non-adjacent levels, AFPN [36] explicitly models long-range interactions between high- and low-level features and employs progressive aggregation, improving cross-scale fusion and small-object performance while maintaining low inference overhead.

Recent studies emphasize lightweight yet selective fusion tailored to dense scenes and edge deployment. HR-FPN [37] preserves high-resolution pathways to enhance tiny-object representation under crowding; in addition, HS-FPN [38] jointly models high-frequency components and spatial perception, further improving cross-scale discrimination in complex backgrounds. Overall, the literature has evolved from static, topology-driven pyramids toward adaptive, attention-enhanced, and computation-efficient fusion: learnable weighting, direction-balanced bidirectional links, and content/attention-aware selection are becoming common practice, providing a viable foundation for dense small-object detection under tight FLOPs and memory budgets.

### 2.3. Detection Head

A detection head maps fused pyramid features to class probabilities and bounding boxes across multiple scales. Early YOLO variants typically placed per-level heads (e.g., P3–P5) on top of FPN/PANet to perform classification and regression in parallel. As applications moved toward real-time and dense scenarios, research increasingly focused on design choices that directly affect optimization stability and latency, including branch organization for prediction, anchor paradigms, and label assignment strategies.

First, decoupled prediction and anchor-free designs have become mainstream. Separating classification from regression alleviates gradient interference and improves localization quality, and—combined with re-parameterized convolutions and efficient normalizations—reduces inference latency (e.g., YOLOv6, YOLOv7) [39,40]. Meanwhile, moving from handcrafted anchors to anchor-free heads, together with IoU-aware losses and improved label assignment, enhances generalization across scales and aspect ratios; recent work (e.g., YOLOv9, YOLOv10) [15,41] further strengthens consistency training and one-to-one decoding to reduce NMS dependence and suppress false positives.

Second, selective attention/dynamic aggregation and end-to-end query-based detection improve separability under dense or crowded conditions. A representative line (e.g., DyHead) [42] integrates attention into head-side aggregation to perform lightweight selection along level, spatial, and channel dimensions, enhancing small-object representation without excessive FLOPs. In parallel, end-to-end query detectors (e.g., RT-DETR and its v2) [24] couple the “head’’ with a transformer decoder, using one-to-one matching and deployable sampling/decoding strategies to simplify post-processing and improve recall and stability in complex backgrounds.

## 3. Method

### 3.1. Overview

In this section, we introduce DBA-YOLO, a lightweight model for dense target detection as shown in Figure 2.

The architecture consists of a backbone, a neck, and a head, in which a modified C2f PA module is integrated to capture normalized object sizes at 80×80, 40×40, and 20×20 resolutions, respectively. The neck uses a streamlined multi-scale feature fusion network, BIMAFPN, which enhances the Path Aggregation Network (PAN) by removing the inefficient “Concat” module and adding four feature fusion modules to focus on the object area and mitigate complex background effects. A modified detection head, DyDHead, is used for localization, classification probability, and object scoring, and it consists of three detection layers representing feature maps of different sizes.

### 3.2. C2f PA Module

The C2f PA module, shown in Figure 3, is an enhanced version of the CSP Bottleneck module with two convolution layers from YOLOv8. The key modification is the introduction of the ParNetAttention attention mechanism, which improves feature selection and multi-scale fusion efficiency. This module consists of three main branches: Channel Attention (SSE), 1 × 1 Convolution, and 3 × 3 Convolution. ParNetAttention enhances feature selection as follows:

Channel Attention (SSE): Global Average Pooling (GAP) is applied to extract global information from the input features. GAP computes the average of each feature map over its entire spatial dimensions (height and width), producing a single value for each feature map. This is followed by a 1 × 1 convolution to capture channel relationships, and the result is passed through a Sigmoid activation to generate attention weights. These weights are then multiplied by the input features to enhance important channels and suppress redundant ones.(1)$$x_{sse}=\sigma \left(f_{1\times 1}\left(\mathrm{GAP}(x)\right)\right)\cdot x$$
where $\mathrm{GAP}\left(x\right)=\frac{1}{H\times W}\sum_{i=1}^{H}\sum_{j=1}^{W}x_{i,j}$ is Global Average Pooling.

A 1 × 1 convolution: The 1 × 1 convolution is used for feature transformation, followed by Batch Normalization (BN) to stabilize the training process.(2)$$x_{1\times 1}=\mathrm{BN}\left(f_{1\times 1}\left(x\right)\right)$$

A 3 × 3 convolution: The 3 × 3 convolution extracts local features, and Batch Normalization (BN) ensures training stability.(3)$$x_{3\times 3}=\mathrm{BN}\left(f_{3\times 3}\left(x\right)\right)$$

Feature fusion and output: The features from the 1 × 1 convolution, 3 × 3 convolution, and Channel Attention (SSE) are fused by element-wise addition, and the output feature map is obtained through SiLU activation:(4)$$y=\mathrm{SiLU}(x_{1\times 1}+x_{3\times 3}+x_{sse})$$

The feature map, after being activated by SiLU, is first divided using the split operation along the channel dimension, reducing the computation and allowing the model to process different subsets of features. Then, multiple Bottleneck modules are applied for deeper feature extraction. Finally, the features from different branches are merged using the concat operation to enhance the feature representation.

This feature fusion strategy enables the model to process information at different scales, while the attention mechanism dynamically adjusts the importance of each channel. This significantly improves feature selection capabilities, balancing accuracy and speed, making it suitable for visual tasks that require both real-time performance and high accuracy.

### 3.3. Multi-Scale Attention Feature Fusion Network

In object detection tasks, objects may appear at different scales due to factors like distance, angle, and occlusion. A single feature extraction method struggles to capture information across these varying scales, leading to potential information loss. The feature pyramid fusion framework addresses this challenge by processing multi-resolution data to detect objects and features at different sizes. We propose the BIMAFPN architecture (Figure 2), which combines a multi-branch auxiliary feature pyramid network (MAFPN) with a bidirectional feature pyramid network (BiFPN). This approach leverages BiFPN’s multi-scale feature fusion capability while incorporating outputs from the spine and neck to maintain optimal shallow information. The advanced auxiliary fusion module, embedded in the neck, conveys diverse gradient information, enhancing the model’s ability to suppress irrelevant background features, ultimately improving detection accuracy and efficiency.

Our lighter-weight neck design, BiFPN, enhances feature selection through weighted cross-scale connectivity. This reduces redundant computations, unifies multi-scale feature channels, and lowers computational costs by fixing the head_channel, minimizing the number of parameters and operations in the detection head. Multi-scale feature maps are analyzed, enabling the model to capture various levels of detail and emphasize the foreground targets for improved detection.

BiFPN realizes multi-layer feature fusion by bidirectional path, where $P_{l}^{\mathrm{out}}$ represents the output feature of level *l*, while $K_{l}$ denotes the set of input levels connected to level *l* (e.g., adjacent levels). Resize refers to upsampling or downsampling operations used to align feature resolutions. Conv represents depthwise separable convolution (including Batch Normalization, BN, and activation functions) to reduce computational complexity. Trainable fusion weights $w_{l,k}$ are constrained to be non-negative (e.g., ReLU) and normalized. The BiFPN formula is as follows:(5)$$P_{l}^{\mathrm{out}}=\mathrm{Conv}\left(\sum_{k\in K_{l}}\frac{w_{l,k}}{\varepsilon +\sum_{k\in K_{l}}w_{l,k}}\cdot \mathrm{Resize}\left(P_{k}^{\mathrm{in}}\right)\right)$$

As shown in Figure 4, the main goal of BiSAF is to combine deep information with features at the same level and high resolution shallow layers within the backbone network to preserve rich localization details and enhance the spatial representation of the network. In addition, we utilize 1×1 convolution to control the number of channels in the shallow information, ensuring that the number of channels in each input is the same, and ensuring that BiFPN operations can be performed without affecting subsequent learning. Let $P_{n-1},P_{n}$ and $P_{n+1}\in R^{H\times W\times C}$ denote the feature mapping at different resolutions, where $P_{n},P_{n}^{\prime }$ and $P_{n}^{\prime \prime }$ denote the feature layers of the backbone network and the two paths of the BIMAFPN. U denotes an upsampling operation.Down denotes a 3×3 downsampling convolution accompanied by a Batch Normalization layer. SiLU denotes a SiLU function, C denotes a 1×1 convolution controlling the number of channels. The output after applying BiSAF is as follows:(6)$$P_{n}^{\prime }=\mathrm{BiFPN}\;\left(\mathrm{SiLU}\;\left(C\;\left(\mathrm{Down}(P_{n-1})\right)\right),\;\mathrm{SiLU}\;\left(C(P_{n})\right),\;U\left(P_{n+1}^{\prime }\right)\right)$$

To improve the interactive use of feature layer information, we integrate the BiAAF module into the deeper layers of the BIMAFPN for multi-scale feature fusion. Specifically, Figure 5 shows the AAF connections in $P_{n}^{\prime }$, which aggregate information from the shallow high-resolution layer $P_{n+1}^{\prime }$, the shallow low-resolution layer $P_{n-1}^{\prime }$, the sibling shallow layer $P_{n}^{\prime }$, and the previous layer $P_{n-1}^{\prime }$. This enables the final output layer to merge data from four distinct layers, significantly enhancing detection performance of medium-sized targets. According to the traditional FPN single-path architecture, we assume that the initial bootstrap information is already embedded in the shallow layers of the BIMAFPN. Therefore, we equalize the number of channels in each layer to ensure that the model obtains different outputs. The outputs after applying BiAAF are as follows:(7)$$P_{n}^{\prime \prime }=\mathrm{bifpn}\left(\mathrm{Down}\left(P_{n-1}^{\prime }\right),\mathrm{Down}\left(P_{n-1}^{\prime \prime }\right),P_{n}^{\prime },U\left(P_{n+1}^{\prime }\right)\right)$$

### 3.4. DyDHead Schematic

The improved detection head DyDHead, as schematically shown in Figure 6, consists of three main modules: scale-aware attention, spatial-aware attention, and task-aware attention. The input tensor to these modules has three key dimensions: L represents the number of levels in the feature pyramid, corresponding to different resolutions of the feature map; S represents the spatial dimension (height and width), i.e., the number of spatial locations in the feature map; and C represents the channel dimension, i.e., the number of feature channels at each spatial location. Each attention module operates on one of these dimensions: scale-aware attention works on L (levels), spatial-aware attention operates on S (spatial locations), and task-aware attention focuses on C (channels). This separation allows the model to focus on different aspects of the feature map, improving detection efficiency and accuracy.

Given a feature tensor $F\in R^{L\times S\times C}$, the general formula for applying self-attention is(8)W(F)=π(F)·F
where π(·) is an attention function, but using a fully connected layer for self-attention is computationally too expensive, so we convert the attention function into three consecutive attentions:(9)$$W\left(F\right)=\pi_{C}\left(\pi_{S}\left(\pi_{L}\left(F\right)\cdot F\right)\cdot F\right)\cdot F$$
where $\pi_{L}\left(\cdot \right),\pi_{S}\left(\cdot \right)$, and $\pi_{C}\left(\cdot \right)$ are three different attention functions applied to dimensions L, S, and C. $\pi_{L}\left(\cdot \right)$ corresponds to the scale-aware attention module, dynamically fusing features of different scales according to their semantic importance. The specific process corresponds to the Scale Attn module in the DyDHead detail design in Figure 7.(10)$$\pi_{L}\left(F\right)\cdot F=\sigma \left(f\left(\frac{1}{SC}\sum_{s,c}F\right)\right)\cdot F$$

$\pi_{S}\left(\cdot \right)$ corresponds to a spatially aware attention module based on fusion of features to continuously attend to discriminative regions co-occurring in spatial locations and feature layers. Considering the high-dimensionality of the spatial dimension *S*, we decompose this module into two steps: first, we make the attention learning sparse by using DCNV3, and then we aggregate features across layers at the same spatial location, where DCNV3 is a third-generation version of deformable convolutional networks designed to improve the performance of Convolutional Neural Networks (CNNs) in dealing with changes in the shapes and locations of objects in images. DCNV3 is optimized on the basis of the previous two generations of DCNs by introducing grouping operations and dynamic offsets to further enhance the deformation-awareness of the model.(11)$$\pi_{S}\left(F\right)\cdot F=\frac{1}{L}\sum_{l=1}^{L}\sum_{k=1}^{K}w_{l,k}\cdot F\left(l;p_{k}+\Delta p_{k};c\right)\cdot \Delta m_{k}$$
where *K* is the number of sparsely sampled positions, $p_{k}+\Delta p_{k}$ are the positions offset by a self-learned spatial offset $\Delta p_{k}$ for attending to discriminative regions, and $\Delta m_{k}$ is a self-learned importance scalar at position $p_{k}$.

To enable joint learning and generalize the representation of different objects, $\pi_{C}\left(\cdot \right)$ corresponds to a task-aware attention module that dynamically turns feature channels on and off to support different tasks.(12)$$\pi_{C}\left(F\right)\cdot F=\max\left(\alpha^{1}\left(F\right)\cdot F_{c}+\beta^{1}\left(F\right),\alpha^{2}\left(F\right)\cdot F_{c}+\beta^{2}\left(F\right)\right)$$

$F_{c}$ denotes the feature slice on the *c*-th channel, while ${\left[\alpha 1,\alpha 2,\beta 1,\beta 2\right]}^{T}=\theta \left(\cdot \right)$ is a hyperfunction for learning the control activation threshold. In the implementation of the function θ(·), the L×S dimensions are first pooled globally on average to reduce the dimensionality, then two fully connected layers and a normalization layer are used, and finally an offset Sigmoid function is applied to normalize the output to the range [−1,1]. The exact process corresponds to the Task Attn module in the DyDHead detailed design in Figure 7.

### 3.5. Loss Function

We follow the default YOLOv10 recipe for losses—CIoU for box regression, Distribution Focal Loss (DFL) for distributional regression, and BCEWithLogits with focal modulation for classification—without changing the default loss weights. The total loss is the weighted sum(13)$$L=\lambda_{\mathrm{box}}L_{\mathrm{box}}+\lambda_{\mathrm{cls}}L_{\mathrm{cls}}+\lambda_{\mathrm{dfl}}L_{\mathrm{dfl}},$$
where $\lambda_{\mathrm{box}},\lambda_{\mathrm{cls}},\lambda_{\mathrm{dfl}}$ follow the baseline’s dynamic weighting strategy for joint optimization, and task-aligned assignment (TAL) [43] is adopted to reduce classification–localization misalignment.

For box regression, we use the Complete IoU loss (CIoU) [44]:(14)$$L_{\mathrm{CIoU}}=1-\mathrm{IoU}+\frac{\rho^{2}\left(b,b^{\mathrm{gt}}\right)}{c^{2}}+\alpha v,$$(15)$$v=\frac{4}{\pi^{2}}\;{\left(\arctan\frac{w^{\mathrm{gt}}}{h^{\mathrm{gt}}}-\arctan\frac{w}{h}\right)}^{\;2},\;\alpha =\frac{v}{(1-\mathrm{IoU})+v}.$$
where ρ(·,·) is the Euclidean distance between box centers, *c* is the diagonal of the minimum enclosing box, and w,h (resp. $w^{gt},h^{gt}$) are the predicted (resp. ground-truth) width and height.

To obtain finer localization at fractional coordinates, we adopt Distribution Focal Loss (DFL) [45]:(16)$$L_{\mathrm{dfl}}=-\sum_{i=0}^{n}\left[\left(y_{i+1}-y^{gt}\right)\log P\left(y_{i}\right)+\left(y^{gt}-y_{i}\right)\log P\left(y_{i+1}\right)\right],$$
where $y^{gt}$ is the continuous target, $y_{i}$ and $y_{i+1}$ are adjacent bins, and $P\left(y_{i}\right),P\left(y_{i+1}\right)$ are their predicted probabilities; we follow the baseline binning strategy (finer near small-object ranges).

For classification, BCEWithLogits with focal modulation [46] is used to handle class imbalance under multi-label supervision:(17)$$L_{\mathrm{cls}}=-\frac{1}{N}\sum_{k=1}^{N}\alpha_{t}{\left(1-p_{t}\right)}^{\gamma }\left[y_{k}\log\sigma \left(p_{k}\right)+\left(1-y_{k}\right)\log\left(1-\sigma (p_{k})\right)\right],$$
where $p_{k}$ is the logit for sample *k*, σ(·) is the Sigmoid, and $p_{t}=\sigma \left(p_{k}\right)$ if $y_{k}=1$ (else $p_{t}=1-\sigma \left(p_{k}\right)$); $\alpha_{t}$ is a class-balance factor and γ is the focusing parameter. We follow the baseline’s curriculum schedule for γ over training epochs *T*.

Following TAL [43], the training target confidence couples localization and classification quality via(18)$$\mathrm{Score}={\mathrm{IoU}}^{\;\beta }\cdot {\mathrm{ClsScore}}^{\;1-\beta },$$
where β∈[0,1] is a balance factor (default 0.5).

## 4. Experiments

### 4.1. Dataset

The experiment utilized SKU-110K [5] and a custom cigarette pack dataset. SKU-110K comprises 11,762 images with 173,678 instances, collected from diverse supermarket stores. The dataset underwent a split of 8821 training samples and 2941 test samples. The self-built cigarette package dataset includes over 1000 real images captured in various cities, weather conditions, and times of day, each sized at 960 pixels × 1280 pixels. To augment diversity and prevent overfitting, 50 natural scene images without cigarette packages were included as negative samples. Labeling was performed using labelImg, resulting in 1073 images, 50,173 marker samples, with 887 images in the training set and 186 samples in the test set. As the original cigarette package images are proprietary, mosaic processing was applied to all displayed faces for privacy reasons, as illustrated in Figure 8.

### 4.2. Experimental Environment and Parameter Settings

The experiment used Python 3.10.14, CUDA 12.1, RTX 4090D GPU (24 GB; NVIDIA Corporation, Santa Clara, CA, USA), and Xeon 8474C CPU (Intel Corporation, Santa Clara, CA, USA). The input image sizes for SKU-110K and cigarette packet datasets were 640×640 and 1280×1280, respectively, and mosaic data enhancement was used to increase diversity. Training involves 100 epochs for SKU-110K and 300 epochs for the cigarette packet dataset to achieve optimal weights. Stochastic Gradient Descent (SGD) with momentum of 0.937 and weight decay coefficient of 0.0005 is used for gradient updating. The initial learning rate is $10^{-2}$ and the final learning rate is $10^{-4}$ with a batch size of 16.

### 4.3. Evaluation Metrics

The experiment uses evaluation indicators in the field of deep learning object detection, such as precision, recall, and average precision (AP). The formulas are as follows:(19)$$\mathrm{Recall}=\frac{TP}{TP+FN}$$(20)$$\mathrm{Precision}=\frac{TP}{TP+FP}$$(21)$$AP=\int_{0}^{1}P\left(R\right)dR$$
where TP represents the number of positive samples correctly predicted as positive, FP represents the number of positive samples incorrectly predicted as positive, and FN represents the number of positive samples incorrectly predicted as negative. *P* represents precision, and *R* represents recall rate. We used the same evaluation method as COCO [47], which reported the mean average precision (mAP) when IoU = 0.5:0.05:0.95 (The IoU ranges from 0.5 to 0.95 with a step size of 0.05). In addition, we report the $AP_{50}$ (IoU = 0.5) and the $AP_{75}$ (IoU = 0.75).

### 4.4. Result

To further investigate the effectiveness and practical deployability of DBA-YOLO for cigarette package target detection, we conduct systematic experiments on the SKU-110K dataset. DBA-YOLO achieves consistent improvements over YOLOv10n in mAP and $AP_{75}$; meanwhile, compared with the transformer-based RT-DETR-R18, it attains higher accuracy with fewer parameters, reflecting a more favorable accuracy–efficiency trade-off. To comprehensively cover the baseline spectrum for shelf-style dense small-object scenarios, we also include a specialized dense detector—DDQ R-CNN R50 and perform side-by-side evaluation under a unified protocol (640 × 640, single-scale, no TTA); the results show that DBA-YOLO, while remaining lightweight, exhibits robust advantages in both accuracy and efficiency (see Table 1). From the overall comparison in Table 1, DBA-YOLO leads among lightweight models (n/tiny tier), showing more stable localization at the stricter IoU threshold ($AP_{75}$); it also maintains comparatively low parameter counts and GFLOPs while matching or surpassing some larger “s” series models in accuracy. These results collectively indicate that DBA-YOLO is feasible for deployment in resource-constrained scenarios.

Figure 9 shows the trend of the results of different models on the same dataset. The experimental results demonstrate that our method consistently outperforms all compared YOLO variants across both mAP@0.5 and mAP@0.5:0.95 metrics. Specifically, our model achieves faster convergence, higher overall detection accuracy, and greater stability throughout training. Notably, in the more rigorous mAP@0.5:0.95 metric, our approach maintains a clear lead over state-of-the-art lightweight models such as YOLOv8s and YOLOv10n, highlighting its superior ability to localize and classify objects precisely. These advantages indicate the strong effectiveness and robustness of our method in practical object detection tasks.

Table 1 highlights the significant improvements of our proposed method, with a 1.9% increase in mean average accuracy mAP and a 2.6% increase in $mAP_{75}$ compared to the baseline model, along with a 3.6% reduction in the number of parameters. Additionally, our method shows a corresponding increase in mAP and $mAP_{75}$ compared to other YOLO series models. While there is a corresponding increase in the number of parameters in YOLOv8s and YOLOv10s, the results are similar. However, YOLOv8s and YOLOv10s exhibit a significant increase in the number of parameters and GFLOPs. The visual representations in Figure 10 demonstrate the improvement of bounding box fitting on the SKU-110K dataset, which significantly reduces the overlap and distinguishes the boundaries between similar objects. The edge regions are better recognized, indicating that our model performs well in accurate bounding box localization, especially for densely distributed objects.

In order to gain further insight into the effectiveness of the DBA-YOLO model in cigarette package target detection and its real possibility of practical application, this paper conducts experiments on the cigarette package dataset, and the results of the experiments are shown in Table 2. Table 2 shows that our model achieves a significant gain of 1.4% in terms of mAP. A significant gain of 1.2% is also realized for $AP_{75}$. Moreover, compared with the transformer-based RT-DETR-R18, our approach delivers higher accuracy with far fewer parameters, mAP
81.0 vs. 79.4
(+1.6), $AP_{50}$
99.4 vs. 99.3
(+0.1), and $AP_{75}$
93.9 vs. 93.1
(+0.8), while reducing parameters from $198.73\times 10^{5}$ to $26.16\times 10^{5}$
(≈86.8%↓). These results further support the practicality of DBA-YOLO for deployment in resource-constrained scenarios.

Figure 11 shows the object detection results on the cigarette package dataset. It can be seen that our model can accurately localize cigarette packages in complex backgrounds. Compared with other object detection algorithms, our model outperforms YOLOv10n by 1.4% in terms of $AP_{75}$, is 1.3% higher than YOLOv11n, and is 1.4% higher than YOLOv12n, which is a significant improvement in performance. Furthermore, comparing to YOLOv10, the improved adaptability will allow deployment of the model in a variety of environments as the model parameters are significantly reduced. It can be transferred to mobile devices such as law enforcement equipment while maintaining reasonable detection accuracy, thus enabling real-time detection.

As shown in Figure 12b,c, the reflected objects from the YOLOv10 glass mirror are recognized as detected, while the DBA-YOLO is unaffected by interference and obtains accurate recognition results. We attribute the improvement to the optimized detection head, which enables the model to focus on cigarette packages and mitigates background interference. Overall, DBA-YOLO achieves the best performance on cigarette package detection, surpassing state-of-the-art baselines in mAP and $AP_{75}$.

Figure 13 shows the comparison of the inference heat maps of YOLOv10 and DBA-YOLO models on the cigarette packet dataset and SKU-110 dataset. In heat map (b), YOLOv10 is weak in capturing local features (e.g., edges) of the target, and the heat map reflects a less precise region, while DBA-YOLO’s heat map (c) significantly reduces the interference of the background and the character’s arm, and the model focuses more on local features (e.g., text and edges) of the target and is able to discriminate the characteristics of the target region in more detail. In heat map (e), although YOLOv10 pays better attention to the box as a whole, some of the hotspots are distributed in regions unrelated to the target (e.g., character’s arms, background edges), and there is a certain amount of distraction; in contrast, DBA-YOLO’s heat map (f) pays more attention to the target region (e.g., the box that is held), with more precise alignment with the edges of the target object, less interference from the background, and a better ability to exclude irrelevant information. In heat map (h), YOLOv10’s hotspot mainly focuses on the center part of the goods on the shelves, but some areas receive less attention, so it fails to completely cover all goods, especially some smaller or inconspicuous items, resulting in missed detection; in contrast, DBA-YOLO’s heat map (i) covers more goods, including small or edge objects that may be overlooked by YOLOv10, with more comprehensive detection, which reduces the possibility of missed detection, and the heat map is more uniform and covers a wider range of items, especially for small and dense objects. Overall, DBA-YOLO outperforms YOLOv10 in terms of target area focus, localized feature capture capability, and comprehensiveness.

In addition to the experiments conducted on the SKU-110K and our custom cigarette package dataset, we further validated the performance of DBA-YOLO on the VisDrone-2019 dataset [48]. VisDrone-2019 is a large-scale UAV imagery dataset featuring rich and densely distributed object detection scenarios, making it particularly suitable for detecting small and dense objects. This dataset includes aerial images captured in various scenes and environments, containing numerous objects of different scales, densities, and partial occlusions.

VisDrone-2019 covers aerial scenes with crowded, small, and partially occluded objects under complex backgrounds. On this benchmark, DBA-YOLO reaches 38.5% mAP@0.5 and 23.4% ${\mathrm{AP}}_{75}$, showing consistent gains over strong baselines (Table 3).

This paper also evaluates the effectiveness of different detection heads to demonstrate the feasibility of DyDHead, as shown in Table 4. All comparisons use the same backbone (YOLOv10n) and neck, the same input size (640), and identical schedule/augmentations; only the detection head differs. In the comparison across multiple heads, DyDHead achieves better detection performance. Although its parameter count may increase relative to some alternatives, the accuracy gains are larger. Compared with SEAMHead, RSCD, TADDH, and LSCD, the metrics mAP, $AP_{50}$, and $AP_{75}$ are substantially improved, with a modest parameter increase in certain settings. In practice, DyDHead produces fewer false detections; relative to MultiSEAMHead, it uses fewer parameters while improving accuracy. These results indicate that DyDHead extracts target-region features more precisely, exploits multi-scale information more efficiently, and strengthens small-object detection—particularly under complex backgrounds.

This paper also evaluates the effectiveness of different neck networks to demonstrate the feasibility of BIMAFPN, as shown in Table 5. All comparisons use the same backbone (YOLOv10n) and head, the same input size (640), and identical schedule/augmentations; only the neck differs. In comparisons across multiple necks, BIMAFPN achieves better detection performance while reducing the parameter count relative to most alternatives. Against RCSOSA, mAP and $AP_{75}$ are comparable, and BIMAFPN uses roughly half the parameters. Compared with other necks, BIMAFPN not only requires fewer parameters but also yields higher mAP, $AP_{50}$, and $AP_{75}$, indicating that the BIMAFPN module more efficiently exploits multi-scale features and strengthens small-object detection. This is particularly useful for detectors that are sensitive to complex backgrounds.

### 4.5. Ablation Study and Discussion

By optimizing the performance of the proposed DBA-YOLO model in terms of detection accuracy and speed in the SKU-110K dataset, we select YOLOv10 as the baseline model and analyze the performance of each component, and the experimental results are shown in Table 6.

#### 4.5.1. Effectiveness of BIMAFPN Module

After replacing the original YOLOv10 neck with the proposed BIMAFPN module, the number of parameters is significantly reduced. The SAF (Shallow-Assisted Fusion) mechanism preserves shallow features via bidirectional connections, improving the detection of small objects. Meanwhile, the AAF (Advanced-Assisted Fusion) enhances gradient flow at the output stage through multidirectional connections. Additionally, the BiFPN structure introduces cross-scale bidirectional paths and weighted feature fusion, improving detection accuracy and efficiency. As shown in Table 6, this modification reduces the number of parameters by approximately 30% while achieving a 0.5% improvement in mAP and a 0.8% gain in $AP_{75}$.

#### 4.5.2. Effectiveness of C2f PA Module

The proposed C2f PA feature fusion module adaptively adjusts the weights of input features based on their relative importance, enhancing feature extraction under complex background conditions. With the integration of an attention mechanism, the model’s representational capacity is further improved. As shown in Table 6, although the parameter count increases slightly, the mAP improves by 0.8%, and $AP_{75}$ improves by 1.2%, demonstrating the effectiveness of the proposed module.

#### 4.5.3. Effectiveness of DyDHead Module

The enhanced DyDHead detection head incorporates dynamic convolution and multiple attention mechanisms to better capture semantic representations in complex scenes. This improves both the robustness and accuracy of the detection. As observed in Table 6, while the number of parameters increases marginally, the mAP increases by 0.9% and $AP_{75}$ improves by 1.4% compared to the baseline, validating the effectiveness of the proposed detection head.

## 5. Discussion

This work presents DBA-YOLO, a lightweight detector tailored for dense-target scenarios under resource-constrained settings. The experiments highlight several contributions that advance the state of the art. Although the absolute gains are modest, DBA-YOLO achieves 1.2–2.6% improvements on mAP/${\mathrm{AP}}_{75}$ with fewer parameters and comparable computing resources, strengthening the accuracy–efficiency trade-off for edge deployment; qualitative evidence (Figure 9, Figure 10, Figure 11 and Figure 12) shows fewer false positives and tighter boxes under crowding/overlap, which global metrics may underestimate. First, integrating C2f PA, BIMAFPN, and DyDHead forms a robust framework for enhanced multi-scale feature extraction and fusion, addressing the long-standing difficulty of small-object detection where conventional models often struggle. In particular, the scale-, spatial-, and task-aware dynamic attentions in DyDHead enable more effective capture of salient cues at varying abstraction levels, yielding strong performance in dense detection tasks. Ablations indicate complementary roles—BIMAFPN for cross-scale aggregation, C2f PA for foreground cues, and DyDHead for separability in dense regions—without inflating depth/width, thereby preserving real-time throughput.

Despite these positive results, several limitations remain. The model performs well in controlled environments, but more dynamic settings—such as extreme illumination or heavy occlusion—can still affect accuracy, motivating further work on robustness. Moreover, while attention improves cross-scale accuracy, there is room to optimize detection in highly textured backgrounds and for very small objects. Concretely, this includes two areas: (i) ultra-small objects near image boundaries are occasionally missed on SKU-110K, and (ii) glass reflections in cigarette imagery can induce spurious responses. Going forward, we aim to strengthen early high-resolution pathways for tiny-object recall and explore reflection-robust attention/de-noising while keeping real-time budgets.

Failure case analysis: Figure 14 provides practitioner-oriented examples. In panel (a), glossy film on cigarette packages introduces specular highlights together with partial hand/edge occlusions; near-duplicate appearances at small scales make NMS more conservative, leading to false negatives (red boxes mark missed instances). In panel (b), shelf-edge blur and extreme density cause boundary objects to become truncated or undersized after resizing, again yielding missed detections. These patterns align with our error logs: most failures occur when targets are tiny, partially visible, or strongly reflective.

Improvement directions: We will proceed along three restrained paths: (1) Data and training: We will adopt augmentations closer to real scenes and moderate sample distribution to mitigate biases from reflection, occlusion, and extreme density. (2) Architecture and representation: We will moderately reinforce early high-resolution features and cross-scale interaction, improving the visibility and separability of tiny and boundary objects while preserving real-time performance. (3) Inference and evaluation: We will refine inference settings for ultra-dense shelves and conduct more systematic error decomposition to continuously locate bottlenecks. Across all paths, we adhere to a “lightweight-first, compute-controlled” principle.

## 6. Conclusions

This paper presents DBA-YOLO, a new approach for dense target detection in complex contexts, utilizing an improved C2f PA module as a backbone feature extraction network. DBA-YOLO is a lightweight network that reduces model parameters and maintains comparable computational complexity, making it suitable for deployment on mobile devices. The BIMAFPN, as a multi-scale attentional feature fusion network, enhances feature extraction and improves detection accuracy. Replacing the original detection head with DyDHead helps the model perform better in complex scenarios. Validation on real cigarette package datasets proves the practical feasibility of DBA-YOLO, which achieves excellent performance on SKU-110K and cigarette package datasets compared with YOLOv10n. With ∼3.6% fewer parameters, DBA-YOLO improves mAP by 1.9% and ${\mathrm{AP}}_{75}$ by 2.6% on SKU-110K. On the cigarette package dataset, it achieves an mAP of 81.0% (+1.6), ${\mathrm{AP}}_{50}$ of 99.4%, and ${\mathrm{AP}}_{75}$ of 93.9%. These results satisfy the detection requirements of complex scenarios. The experimental results confirm that DBA-YOLO outperforms existing dense target detection models in terms of accuracy and local bounding box prediction in complex environments.

## Figures and Tables

**Figure 1 jimaging-11-00345-f001:**
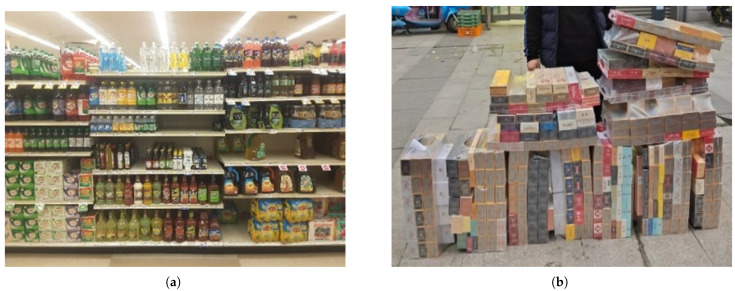
Examples of dense imagery: (**a**) retail shelf display; (**b**) cigarette packages.

**Figure 2 jimaging-11-00345-f002:**
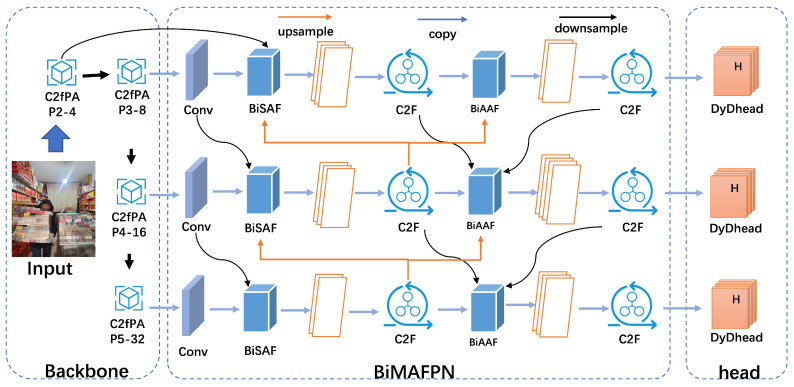
The architecture of the DBA-YOLO method.

**Figure 3 jimaging-11-00345-f003:**
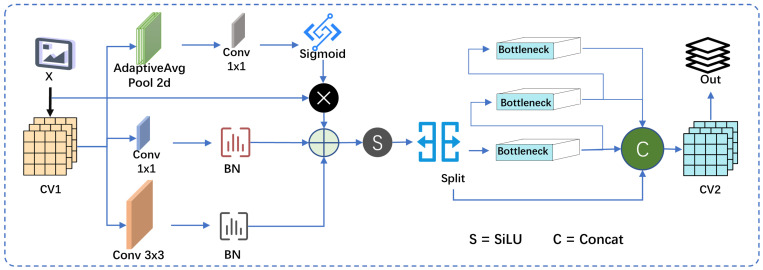
C2f×PA module.

**Figure 4 jimaging-11-00345-f004:**
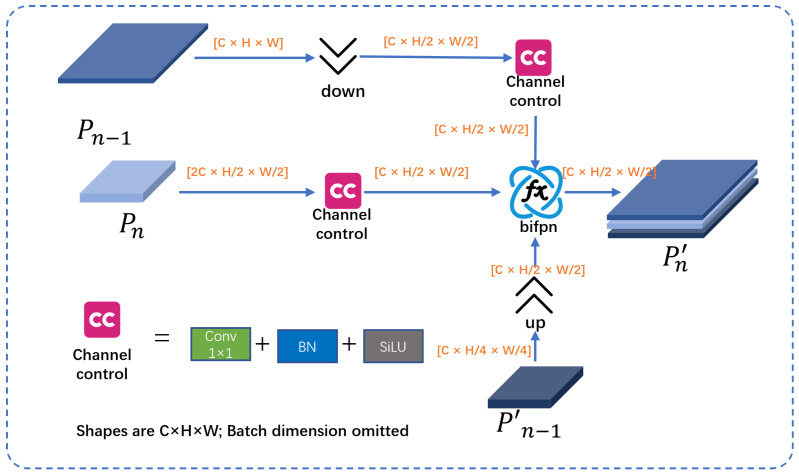
BiSAF structure.

**Figure 5 jimaging-11-00345-f005:**
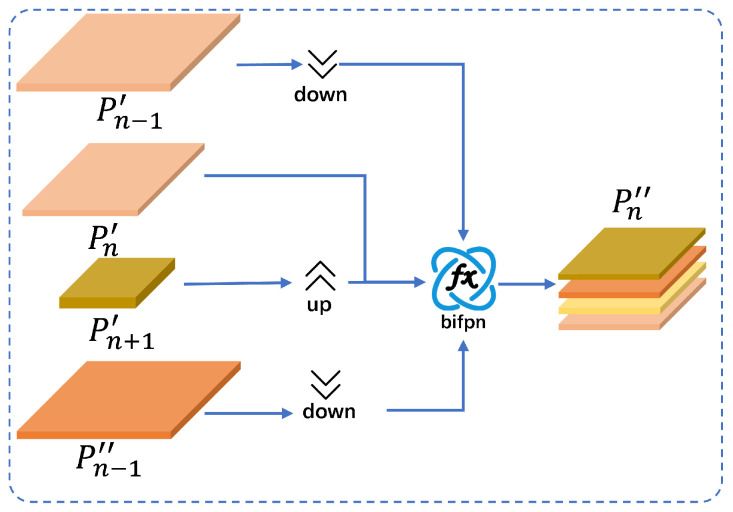
BiAAF structure.

**Figure 6 jimaging-11-00345-f006:**
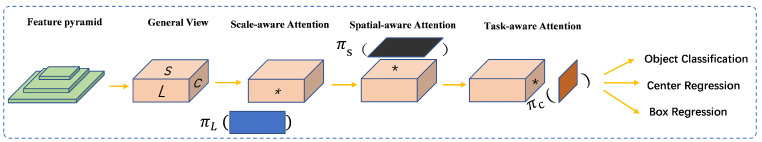
DyDHead schematic. The asterisk (*) marks this face only.

**Figure 7 jimaging-11-00345-f007:**
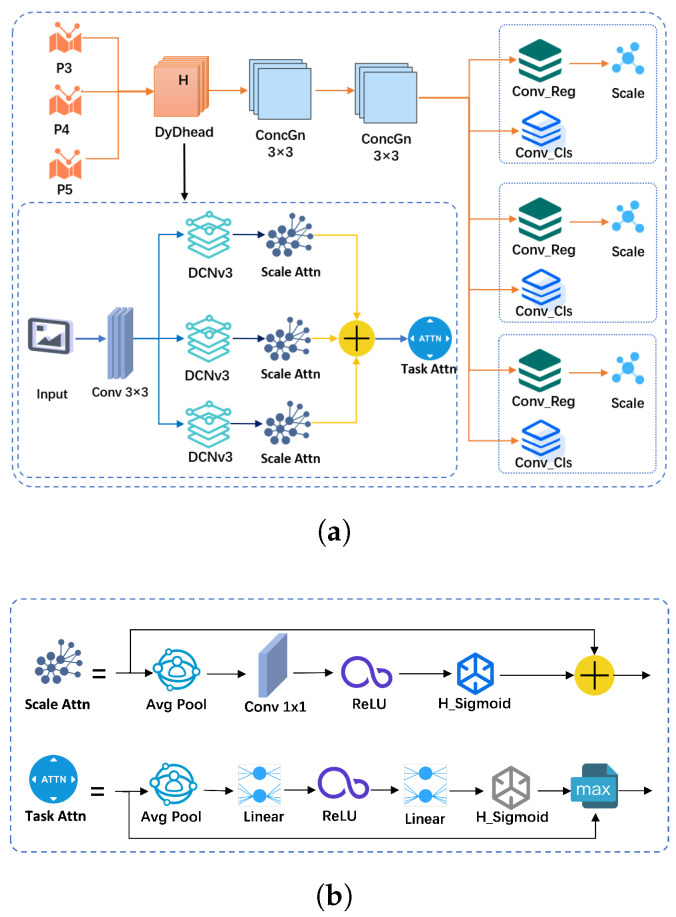
DyDHead structure. (**a**) Detailed structure. (**b**) Scale and task attention modules.

**Figure 8 jimaging-11-00345-f008:**
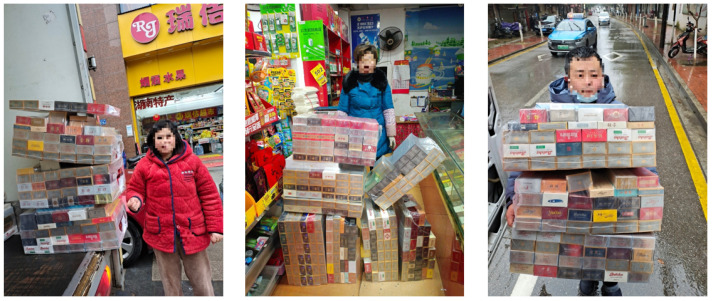
Cigarette package images.

**Figure 9 jimaging-11-00345-f009:**
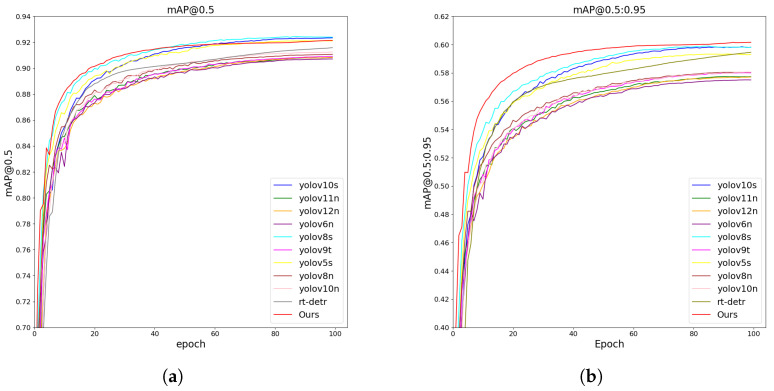
Result curves on SKU-110K: (**a**) The curve on the SKU-110K with mAP@0.5. (**b**) The curve on the SKU-110K dataset with mAP@0.5:0.95.

**Figure 10 jimaging-11-00345-f010:**
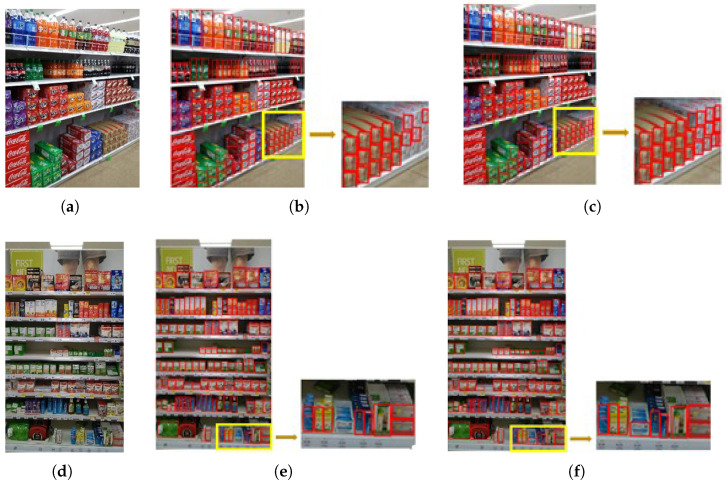
SKU-110K results. (**a**,**d**) Full images; the yellow box marks the region of interest (ROI) used for zoomed comparison. (**b**,**e**) Zoomed ROI with YOLOv10 predictions. (**c**,**f**) Zoomed ROI with DBA-YOLO predictions.

**Figure 11 jimaging-11-00345-f011:**
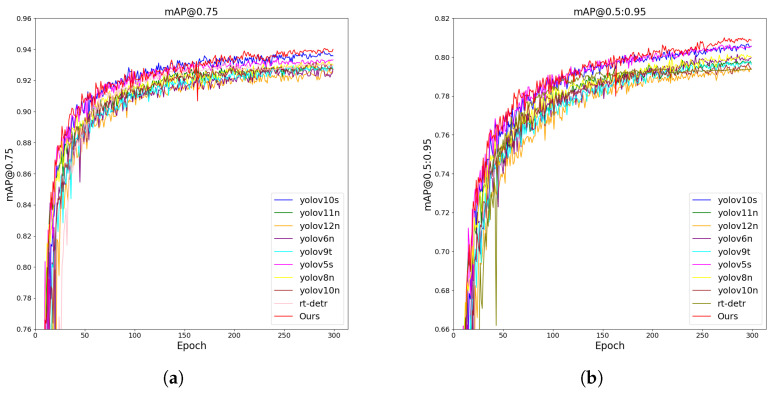
Result curves on cigarette package images: (**a**) The curve on cigarette images with mAP@0.75. (**b**) The curve on cigarette images with mAP@0.5:0.95.

**Figure 12 jimaging-11-00345-f012:**
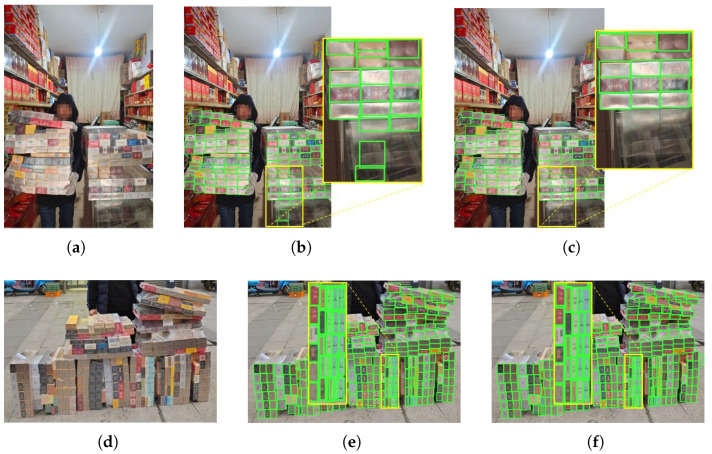
Cigarette package images results. (**a**,**d**) Original images; (**b**,**e**) YOLOv10 predictions; (**c**,**f**) DBA-YOLO predictions. Yellow boxes mark the region of interest (ROI) used for side-by-side comparison; green boxes indicate ground-truth bounding boxes.

**Figure 13 jimaging-11-00345-f013:**
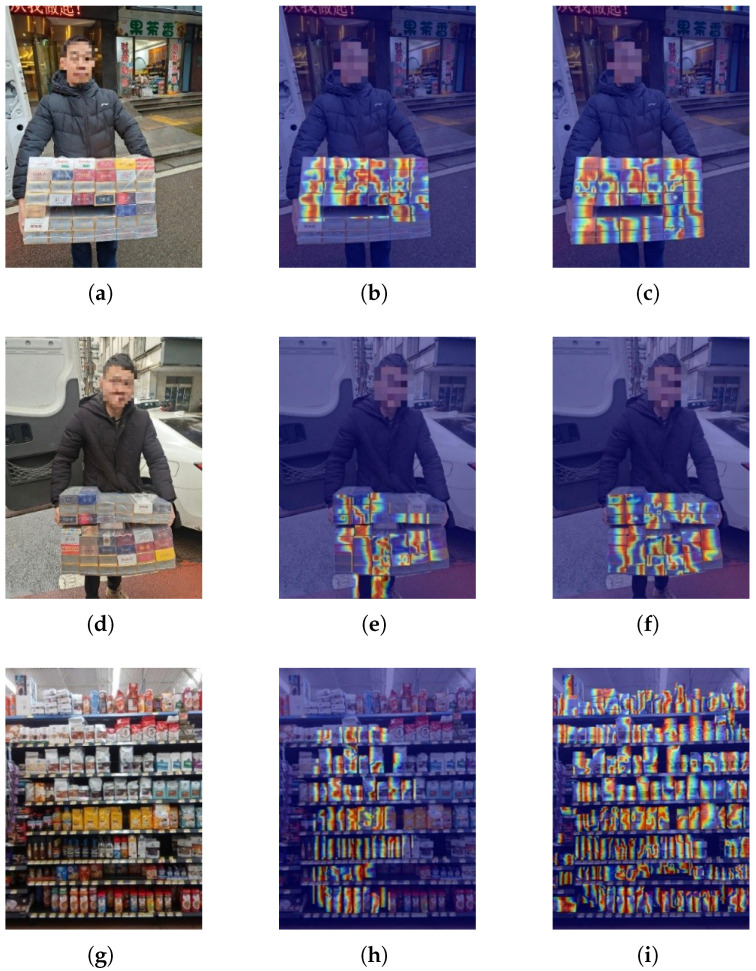
Thermograms (**a**,**d**,**g**) are the original images, (**b**,**e**,**h**) are the thermograms of YOLOv10, and (**c**,**f**,**i**) are the thermograms of DBA-YOLO.

**Figure 14 jimaging-11-00345-f014:**
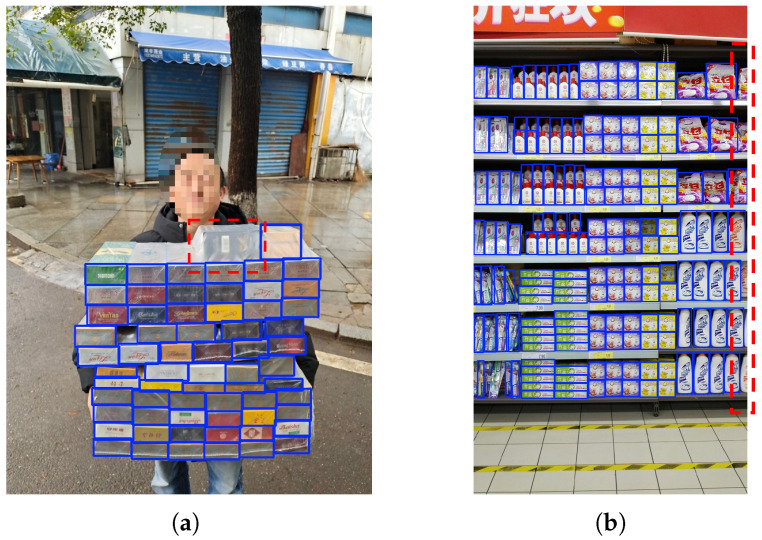
Failure cases. Red dashed boxes indicate missed detections (false negatives). (**a**) Cigarette package example images; (**b**) SKU-110K example images.

**Table 1 jimaging-11-00345-t001:** SKU-110K result.

Dataset	Model	mAP (%)	${\mathrm{AP}}_{50}$ (%)	${\mathrm{AP}}_{75}$ (%)	Params (×$10^{5}$)	GFLOPs
SKU-110K	yolov5n	56.3	88.9	64.1	25.09	7.2
	yolov5s	58.1	90.2	66.8	91.23	24.0
	RTMDet-Tiny	40.1	62.5	46.9	48.73	8.03
	yolov6n	56.4	89.0	64.0	42.38	11.9
	yolov8n	57.0	89.3	65.1	30.11	8.2
	yolov8s	58.7	90.5	67.8	113.60	28.6
	yolov9t	57.0	89.1	65.2	20.06	7.8
	yolov10n	56.9	89.6	65.0	27.07	7.3
	yolov10s	58.7	90.6	67.7	80.67	24.8
	yolov11n	56.4	88.6	64.4	25.90	6.4
	yolov12n	56.6	89.1	64.5	25.68	6.5
	RT-DETR-R18	58.2	89.6	66.9	198.73	56.9
	DDQ R-CNN	38.1	90.3	57.6	632.80	50.2
	OURS	58.8	90.6	67.6	26.16	7.9

**Table 2 jimaging-11-00345-t002:** Cigarette packet result.

Dataset	Model	mAP/%	${\mathrm{AP}}_{50}/\%$	${\mathrm{AP}}_{75}/\%$	Params/$10^{5}$
Cigarette packet	yolov5n	79.9	98.3	92.8	25.09
	yolov5s	80.6	99.1	93.3	91.23
	yolov6n	80.1	98.3	92.6	42.38
	yolov8n	80.2	98.6	93.1	30.11
	yolov8s	81.0	99.2	93.5	113.6
	yolov9t	79.7	98.2	92.8	20.06
	yolov10n	79.6	98.7	92.7	27.07
	yolov10s	80.7	99.1	93.7	80.67
	yolov11n	79.7	98.6	92.9	25.90
	yolov12n	79.6	98.8	92.7	25.68
	RT-DETR-R18	79.4	99.3	93.1	198.73
	Ours	81.0	99.4	93.9	26.16

**Table 3 jimaging-11-00345-t003:** Comparison of object detection performance on VisDrone-2019 validation and test datasets.

Dataset	Model	P	R	mAP/%	mAP50/%	mAP75/%	Params/$10^{5}$
Visdrone-val	yolov10n	0.458	0.35	20.3	35.2	20.4	27.07
	yolov11n	0.441	0.34	19.5	33.7	19.3	25.9
	yolov12n	0.44	0.335	19.3	33.1	19.1	25.68
	OURS	0.504	0.369	22.9	38.5	23.4	26.16
Visdrone-test	yolov10n	0.386	0.302	14.8	27.1	14.4	27.07
	yolov11n	0.393	0.296	15.1	27.1	15.1	25.9
	yolov12n	0.39	0.292	15.2	27	15.2	25.68
	OURS	0.436	0.311	17.6	30.7	17.8	26.16

**Table 4 jimaging-11-00345-t004:** Model comparison on SKU-110 dataset.

Head	mAP (%)	AP50 (%)	AP75 (%)	Params ($\times 10^{5}$)	GFLOPS
Ours	57.8	90.0	66.4	27.8	7.7
SEAMHead [49]	56.6	89.3	64.4	25.2	7.3
TADDH [50]	56.5	89.7	64.8	19.9	8.4
MultiSEAM [51]	56.6	89.3	64.5	67.3	9.3
LSCD [52]	56.9	89.5	65.1	19.5	6.2
RSCD [53]	55.2	87.5	62.8	20.5	6.5

**Table 5 jimaging-11-00345-t005:** Model comparison on SKU-110 dataset.

Neck Networks	mAP (%)	AP50 (%)	AP75 (%)	Params ($\times 10^{5}$)	GFLOPS
Ours	57.4	89.7	65.8	18.9	6.3
bifpn [54]	57.1	89.3	65.1	17.2	6.0
slimneck [55]	56.6	89.3	64.3	23.9	5.9
goldyolo [56]	55.9	88.6	64.8	53.9	8.9
ASF [57]	57.1	89.6	65.1	23.0	6.9
CFPT [58]	56.3	89.6	63.9	18.9	6.4
RCSOSA [59]	57.4	90.0	65.6	41.1	15.3
GFPN [60]	57.0	89.5	65.2	33.2	7.0
EfficientRepBiPAN [61]	56.8	89.5	64.8	27.3	6.8
HSFPN [62]	56.0	88.6	63.5	19.3	6.7

**Table 6 jimaging-11-00345-t006:** Ablation experiments for each module in the DBA-YOLO model.

C2f PA	BIMAFPN	DyDHead	mAP/%	${\mathrm{AP}}_{75}$/%	Params ($\times 10^{5}$)	GFLOPs
–	–	–	56.9	65.0	27.0	7.3
✓	–	–	57.7	66.2	27.7	8.4
–	✓	–	57.4	65.8	18.9	6.3
–	–	✓	57.8	66.4	27.8	7.7
✓	✓	–	57.9	66.4	24.0	7.4
✓	–	✓	58.2	66.8	28.8	8.7
–	✓	✓	58.0	66.7	25.4	7.3
✓	✓	✓	58.8	67.6	26.1	7.9

## Data Availability

The data that support the findings of this study are available from the corresponding author upon reasonable request.

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
