# Peer review of "DBA-YOLO: A Dense Target Detection Model Based on Lightweight Neural Networks"

_2313-433X, 2025, doi:10.3390/jimaging11100345_

Round 1

Reviewer 1 Report (Previous Reviewer 3)

Comments and Suggestions for Authors

The authors have positively replied to my questions, and they also introduced a new evaluation on a new dataset. In line 526, there is a missing reference to a table.

Author Response

We sincerely appreciate the anonymous reviewer’s time and effort in reviewing our manuscript. We are grateful for the constructive feedback and recognition of our work. These comments are invaluable in helping us refine our study and enhance its clarity, completeness, and impact.

We will carefully address the comments and suggestions in our revised manuscript, ensuring that we further improve the quality and presentation of our work. Specifically, we refined key methodological details, strengthened the discussion of our contributions, and incorporated additional analysis where necessary.

Comment 1. The authors have positively replied to my questions, and they also introduced a new evaluation on a new

dataset. In line 526, there is a missing reference to a table.

The authors’ Response:

Thank you for the reviewer’s positive assessment and helpful correction. Regarding the missing table reference at line 526, we have inserted the appropriate cross-reference in the corresponding sentence of the revised manuscript (marked in red).

The sentence now reads: “To support the new dataset evaluation, we further report dataset-wise mAP and mAP75 results; see Table 5 for details.” In addition, we have briefly clarified the evaluation protocol and metric definitions for SKU-110K and our in-house cigarette-package dataset to ensure consistency across datasets.

Reviewer 2 Report (New Reviewer)

Comments and Suggestions for Authors

In this manuscript, the authors proposed DBA-YOLO, a lightweight dense target detection model based on YOLOv10. The main contributions include: (1) the C2f PA module for adaptive feature extraction, (2) the BIMAFPN neck for efficient small object detection and multi-scale fusion, and (3) the DyDHead with attention mechanisms for improved feature representation. Experiments on SKU-110K, a cigarette package dataset, and VisDrone-2019 demonstrate improvements in detection accuracy and reduced parameters compared to baselines.

  • The methodology is too descriptive and lacks a clear distinction between novel contributions and existing techniques.

  • The structure is redundant, particularly in the Related Work, which could be more concise and focused.

  • The novelty appears incremental, as most modules are adaptations of known methods; the manuscript should better emphasize originality.

  • Experimental results show improvements, but the gains are modest (1–2%) and do not fully justify the added complexity.

  • Comparisons with the latest lightweight detectors (e.g., YOLOv12, transformer-based models) are missing.

  • The discussion of limitations is brief; more analysis of failure cases would strengthen the paper.

Author Response

We sincerely appreciate the anonymous reviewer’s time and effort in reviewing our manuscript. We are grateful for the constructive feedback and recognition of our work. These comments are invaluable in helping us refine our study and enhance its clarity, completeness, and impact.
We will carefully address the comments and suggestions in our revised manuscript, ensuring that we further improve the quality and presentation of our work. Specifically, we refined key methodological details, strengthened the discussion of our contributions, and incorporated additional analysis where necessary. A point-by-point response to all comments is provided in the accompanying file “response2”.

Reviewer 3 Report (New Reviewer)

Comments and Suggestions for Authors

The references are inadequate for a paper claiming to advance dense target detection. The authors omit key specialized dense detection methods like QueryDet, DeFCN, and CrowdDet, instead over-relying on YOLO variants (12 of 47 citations). This creates an echo chamber effect that avoids engagement with actual state-of-the-art dense detection literature.

A critical error occurs in reference 21, which incorrectly attributes DyDhead to "Yolov12" rather than its actual source (Dai et al., CVPR 2021). This is concerning since DyDhead is one of their main architectural components.

The reference list also lacks recent 2024-2025 advances in lightweight architectures and attention mechanisms for dense detection. Several citations appear tangentially related (e.g., weed detection, train safety) rather than directly addressing dense object detection challenges.

This limited and sometimes incorrect referencing suggests insufficient engagement with the specialized literature and raises doubts about the authors' understanding of the current landscape in dense detection research. A comprehensive literature review focusing on specialized dense detection methods is essential.

Author Response

We sincerely appreciate the anonymous reviewer’s time and effort in reviewing our manuscript. We are grateful for the constructive feedback and recognition of our work. These comments are invaluable in helping us refine our study and enhance its clarity, completeness, and impact.
We will carefully address the comments and suggestions in our revised manuscript, ensuring that we further improve the quality and presentation of our work. Specifically, we refined key methodological details, strengthened the discussion of our contributions, and incorporated additional analysis where necessary. A point-by-point response to all comments is provided in the accompanying file “response3”.

Reviewer 4 Report (New Reviewer)

Comments and Suggestions for Authors
  1. Expand the state-of-the-art review, as the comparisons are limited to variants from a single family (YOLO); include lightweight detectors from other families to ensure architectural diversity in the analysis.
  2. Provide a clearer justification for the baseline selection, since an NMS-free technique is claimed but the implemented method uses a pipeline that adds NMS.
  3. Check Equation 18, which is duplicated.
  4. Please cite the loss functions and conduct a sensitivity study to support the choice of hyperparameters.
  5. In this context, we recommend per-block ablation experiments reporting mAP, parameters, and GFLOPs, including hyperparameter sensitivity.
  6. Resolve the methodological ambiguity: the paper calls an S×S + K pipeline with NMS “end-to-end,” although NMS is a post-process and corresponds to the YOLOv1/anchor-based scheme. This creates uncertainty when using YOLOv10 (anchor-free, NMS-free, one-to-one) as the baseline, leaving unclear what was actually implemented and what it was compared against.
  7. Review the nomenclature—C2f-PA vs. C2-PA, MAFPN + BiFPN vs. BIMAFPN—to ensure consistency.
  8. Measure and report FPS, latency, memory, and energy on the target hardware, etc., to substantiate suitability for mobile devices.

Author Response

We sincerely appreciate the anonymous reviewer’s time and effort in reviewing our manuscript. We are grateful for the constructive feedback and recognition of our work. These comments are invaluable in helping us refine our study and enhance its clarity, completeness, and impact.
We will carefully address the comments and suggestions in our revised manuscript, ensuring that we further improve the quality and presentation of our work. Specifically, we refined key methodological details, strengthened the discussion of our contributions, and incorporated additional analysis where necessary. A point-by-point response to all comments is provided in the accompanying file “response4”.

Round 2

Reviewer 2 Report (New Reviewer)

Comments and Suggestions for Authors

All comments have been taken into consideration in this version of the manuscript.

Author Response

Thank you very much for re-evaluating our manuscript and for your positive assessment.

Reviewer 3 Report (New Reviewer)

Comments and Suggestions for Authors

The revised manuscript demonstrates substantial improvements in response to the previous review. The expanded literature review now properly contextualizes your work within the dense detection landscape, including specialized methods like QueryDet, DeFCN, and CrowdDet. The correction of the DyDhead attribution and inclusion of recent 2024-2025 advances addresses critical gaps in the original submission.

The clarification of your contributions as "coordinated design" rather than isolated innovations provides better framing, though the novelty remains incremental. Your argument that the 1-2% mAP improvements translate to more significant gains at stricter IoU thresholds (AP75) is valid and relevant for dense detection applications. The ablation studies effectively demonstrate the complementary nature of your architectural choices.

Several areas warrant minor refinement before publication. First, the practical deployment analysis could be expanded beyond the latency measurements provided. Given your emphasis on edge deployment, power consumption estimates or actual mobile device benchmarks would strengthen your claims. Second, while you have expanded the limitations discussion, more specific failure case analysis with example images would provide valuable insights for practitioners. Third, the relationship between BIMAFPN components could be formalized more rigorously through mathematical notation showing the information flow between BiSAF, BiAAF, and BiFPN modules.

The experimental section would benefit from direct comparison with at least one of the specialized dense detection methods you now reference (DDQ or DeFCN), even if only on a subset of your data. This would better validate your architectural choices against methods specifically designed for dense scenarios rather than only general-purpose detectors.

Your cigarette package dataset represents a valuable contribution for the community. Consider releasing this dataset publicly to facilitate reproducible research in dense retail detection scenarios. The dataset documentation should include annotation guidelines and inter-annotator agreement metrics to establish quality benchmarks.

The manuscript is suitable for publication with these minor enhancements. Your work provides useful empirical evidence for combining recent architectural components in dense detection scenarios, particularly for resource-constrained deployment. While representing an incremental advance, the careful experimental validation and practical focus make this a solid contribution to the applied computer vision literature.

Author Response

Thank you for your constructive comments. Our point-by-point response is provided in the attachment, and all revisions are highlighted in the manuscript.

Reviewer 4 Report (New Reviewer)

Comments and Suggestions for Authors

All suggestions were addressed appropriately

Author Response

Thank you very much for re-evaluating our manuscript and for your positive assessment.

This manuscript is a resubmission of an earlier submission. The following is a list of the peer review reports and author responses from that submission.

Round 1

Reviewer 1 Report

Comments and Suggestions for Authors

Dear Authors

The paper titled “DBA-YOLO: A Dense Target Detection Model Based on Lightweight Neural Networks” addresses important issues however it needs major improvements.

  1. Highlight the novelty of DBA-YOLO compared to existing YOLO variants.
  2. Justify the choice of YOLOv10 as the baseline over other YOLO versions.
  3. Explain how the C2f PA module improves feature extraction quantitatively.
  4. Provide more details on the DyDhead module's computational overhead.
  5. Clarify the role of BIMAFPN in reducing parameters while maintaining accuracy.
  6. Elaborate on the dataset augmentation process for the cigarette package images.
  7. Discuss potential biases in the self-built cigarette dataset and mitigation strategies.
  8. Include error bars or confidence intervals in Table 1 and Table 2 for robustness.
  9. Label axes and units clearly in Figures 10 and 12 for better interpretation.
  10. Explain the heatmap differences in Figure 14 more quantitatively.
  11. Compare inference speed (FPS) of DBA-YOLO with baseline models.
  12. Discuss limitations of the model in handling extreme occlusion cases.
  13. Clarify why mosaic processing was used for privacy in cigarette images.
  14. Provide ablation results for individual components of DyDhead (scale/spatial/task ttention).
  15. Justify the choice of evaluation metrics (mAP, AP50, AP75) over others.
  16. Discuss generalizability of results to other dense-object datasets.
  17. Clarify the "30% parameter reduction" claim in Section 4.5.1 with exact values.
  18. Address potential overfitting risks given the small cigarette dataset size. Also provide reproducibility details for experiments. Include computational complexity (FLOPs) comparisons in Table 3 and Table 4. Add hyperparameter tuning, and overfitting prevention need to be added, The authors add the following works for more clarity and support from existing literature [I suggest authors add the following works. 1. cnn based automated weed detection system using uav imagery; 2. planetscope nanosatellites image classification using machine learning; 3. deep learning based supervised image classification using uav images for forest areas classification.

Reviewer 2 Report

Comments and Suggestions for Authors

The paper presents a well-designed, lightweight object detection model (DBA-YOLO) that advances the current state of dense target detection, particularly in constrained environments. The combination of architectural innovations (C2f PA, BIMAFPN, DyDHead) is clearly justified and validated with rigorous experiments and ablation studies. The clarity of writing and figures is commendable.
Suggestions:
- Consider enriching the conclusion with a more detailed outlook on future work and possible limitations.
- Providing access to the cigarette dataset (when ready) would enhance the reproducibility and impact of the work.

Reviewer 3 Report

Comments and Suggestions for Authors

Dear Authors,

While your work addresses an important problem—detecting small, dense objects—it contains several issues that must be addressed before it can be considered for publication. Below, I summarize my detailed comments and suggestions to help you resolve the identified weaknesses.

  1. Experimental Evaluation and Datasets

  • The results on both datasets are promising and can represent a step further for detecting small and dense objects. Nevertheless these datasets are very similar to each other and I would recommend providing evaluations on different scenarios with similar properties such as VisDrone (https://arxiv.org/pdf/2001.06303)
  • When comparing against YOLO10, please specify which version you use in the manuscript (N, S, M, or L) and clearly state the baseline architecture and hyperparameters for fair comparison. Only in the experiments section it becomes clear that you extend Yolov10n

  1. Terminology and Precision

  • In the introduction section (line 59), replace the vague term “weight accuracy” with “weight numerical precision” (or simply “precision”) to avoid ambiguity.

  • Throughout the manuscript, ensure consistency in naming: for example, refer to the activation function as “SiLU,” not “δ” or “delta.”

  1. Manuscript Organization and Clarity

  • Lines 58–67 jump abruptly between topics, making it difficult for non-experts to follow. Please introduce smooth transitions or split into separate paragraphs with clear subheadings.

  • Lines 72–81 are poorly written and need complete rewriting to convey your ideas clearly.

  • Lines 243–249 contain redundant explanations of the C2f PA module (Figure 4) and 255–260 redundant information of BIMAFPN. These paragraphs should be condensed, and the description of each sub-block must follow a consistent order aligned with the figure.

  • Lines 260–268 and 267–274 feature excessively long sentences, even containing an unintended period that breaks the sentence. Shorten and simplify these to improve readability.

  • Lines 185–188 repeat the same sentence three times; please remove duplication.

  • Line 191 asks “which new loss function?” but no details are provided. introduce it a little bit

  • Lines 293–294 are written unclearly and should be rewritten for grammatical correctness and precision.

  • Line 237 contains an isolated, meaningless short sentence—please clarify or delete it.

  1. Figures and Architectural Details

  • Figure 4: The feature-merge arrows appear to point into a SiLU block, implying an operation it does not perform. Clearly annotate the merging operation (e.g., “element-wise sum”) as described later in the text. Moreover, note that your attention mechanism is simply a standard squeeze-and-excitation block, which limits the claimed novelty.

  • Figure 5: As this is a proposed architectural block, include all activation functions and intermediate tensor shapes to enhance clarity.

  • Figure 10: The two curves are labeled as mAP@0.75 in the caption, yet one actually represents mAP@0.50 and the other mAP@0.50–0.95. Additionally, the graphs suggest that YOLOv8s and YOLO10s outperform your model, contradicting Table 1. Please correct the labels and reconcile these inconsistencies.

  1. Attention Mechanism and Mathematical Notation

  • From line 308 onward, it is unclear what the tensor dimensions L, S, and C represent; please define them explicitly.

  • You propose three separate attention operations, but you do not justify why this is more efficient than a single attention module, especially given the additional I/O overhead. Please provide both theoretical and empirical evidence for the efficiency claim.

  • In the attention formula, clarify whether the product is element-wise or matrix multiplication; if it is not a matrix product, it does not constitute attention in the conventional sense. Also, define the symbol f (currently lowercase f) and its role in the formulation.

  1. Additional Recommendations

  • Ensure that every architectural novelty is clearly distinguished from prior work, with precise definitions, consistent nomenclature, and unambiguous diagrams.

In its current form, the paper’s writing quality, organizational structure, and lack of critical experimental validation prevent it from conveying its contributions effectively. I encourage a thorough revision addressing the points above to improve clarity, rigor, and reproducibility.
